# The Helix-Loop-Helix motif of human EIF3A regulates translation of proliferative cellular mRNAs

**Marina P. Volegova[1¤], Cynthia Hermosillo[1], Jamie H. D. Cate[1,2,3]***

**1** Department of Molecular and Cell Biology, University of California, Berkeley, CA, United States of America, **2** Department of Chemistry, University of California, Berkeley, CA, United States of America, **3** Molecular Biosciences and Integrated Bioimaging, Lawrence Berkeley National Laboratory, Berkeley, CA, United States of America

¤ Current address: Pediatric Oncology, Dana Farber Cancer Institute, Boston, MA, United States of America
* j-h-doudna-cate@berkeley.edu

**Data Availability Statement:** All data has been deposited in Gene Expression Ombinus (accession number GSE118239).

**Funding:** This work was supported by National Institutes of Health [R01-GM065050, P50

## Abstract

Improper regulation of translation initiation, a vital checkpoint of protein synthesis in the cell, has been linked to a number of cancers. Overexpression of protein subunits of eukaryotic translation initiation factor 3 (eIF3) is associated with increased translation of mRNAs involved in cell proliferation. In addition to playing a major role in general translation initiation by serving as a scaffold for the assembly of translation initiation complexes, eIF3 regulates translation of specific cellular mRNAs and viral RNAs. Mutations in the N-terminal Helix-Loop-Helix (HLH) RNA-binding motif of the EIF3A subunit interfere with Hepatitis C Virus Internal Ribosome Entry Site (IRES) mediated translation initiation *in vitro*. Here we show that the EIF3A HLH motif controls translation of a small set of cellular transcripts enriched in oncogenic mRNAs, including *MYC*. We demonstrate that the HLH motif of EIF3A acts specifically on the 5′ UTR of *MYC* mRNA and modulates the function of EIF4A1 on select transcripts during translation initiation. In Ramos lymphoma cell lines, which are dependent on MYC overexpression, mutations in the HLH motif greatly reduce MYC expression, impede proliferation and sensitize cells to anti-cancer compounds. These results reveal the potential of the EIF3A HLH motif in eIF3 as a promising chemotherapeutic target.

## Introduction

Eukaryotic translation initiation is tightly controlled and its deregulation can lead to a wide variety of disorders, including cancer [1]. During canonical translation initiation, the eukaryotic small (40S) ribosomal subunit first associates with initiation factors eIF1, eIF1A, eIF3, and eIF5, and is subsequently loaded with the eIF2 ternary complex (eIF2-GTP-Met-tRNA$^{Met}_i$). It then binds to the mRNA-bearing eIF4F complex (eIF4A, eIF4G, eIF4E) to begin directional scanning of the 5′ untranslated region (5′ UTR) for the start codon [2]. Distinct from the canonical scanning mechanism, recent evidence indicates that eIF3 –the largest of the translation initiation factors comprised of 12 tightly associated and 1 loosely associated subunits (EIF3A-M) in mammals–regulates alternative pathways of translation initiation. For select cellular mRNAs, eIF3 can either activate or repress translation by interacting with RNA structural

GM102706 to J.H.D.C.]. This work used the Vincent J. Coates Genomics Sequencing Laboratory at UC Berkeley, supported by the National Institutes of Health Instrumentation Grant S10 OD018174. See: https://reporter.nih.gov/ The funders had no role in study design, data collection and analysis, decision to publish, or preparation of the manuscript.

**Competing interests:** The authors have declared that no competing interests exist.

elements in the 5′ UTRs of these mRNAs [3]. Additionally, eIF3 can bind the 5′-cap of mRNAs using EIF3D [4, 5], allowing translation of select transcripts to continue under cellular stress or starvation conditions when eIF4E is inactivated. Finally, eIF3 can also promote translation under stress conditions by binding $m^6$A-methylated 5′ UTRs in a cap-independent manner [6, 7]. In all these cases, the molecular basis for eIF3-dependent translation regulation and its control of gene expression networks remain unclear.

Structural analysis using cryo-electron microscopy (cryo-EM) revealed that the core of eIF3, a five-lobed octameric complex, localizes to the solvent-exposed "backside" of the 40S subunit and spans the mRNA entry and exit channels [8]. In the mammalian 43S pre-initiation complex (PIC), eIF3 subunits EIF3A and EIF3C directly contact the 40S subunit [9], as well as participate in interactions with eIF1, eIF1A, eIF2, eIF5, and eIF4F [10, 11], thus coordinating the ordered assembly of the 48S initiation complex. By contrast with canonical initiation, translation of the Hepatitis C Virus (HCV) genomic RNA requires an Internal Ribosome Entry Site (HCV IRES) RNA structure in the 5′ UTR, which binds to subunits EIF3A and EIF3C within the eIF3 complex [12]. Cryo-EM structures later revealed how an HCV-like viral IRES displaces eIF3 from binding the 40S subunit, while still binding to eIF3 through subunits EIF3A and EIF3C [13] (Fig 1A). A key functional element of the EIF3A subunit in mediating RNA binding is a predicted N-terminal Helix-Loop-Helix (HLH) motif. The importance of the HLH motif is revealed by *in vitro* translation (IVT) experiments in which mutation of amino acids 36–39 (KSKK > NSEE) disrupted eIF3 binding to the HCV IRES [12]. The triple amino-acid mutation was sufficient to abrogate binding to and subsequent translation of HCV IRES RNA, without disrupting eIF3 complex assembly [12]. These experiments determined that the HLH motif of EIF3A is critical for mediating HCV IRES binding by eIF3, and thus indicated the need to investigate the importance of this motif for cellular mRNA translation.

Beyond the role of EIF3D in binding the mRNA $m^7$G cap [4], the mechanisms responsible for eIF3-mediated regulation of specific cellular mRNAs remain unclear. Structural models for eIF3 bound to 43S and 48S pre-initiation complexes suggest that eIF3 controls the translation of the HCV IRES in a distinct manner compared to cellular transcripts, whether involving canonical scanning or eIF3-dependent regulation of specific transcripts. In addition to its displacement from the 40S subunit by the HCV-like IRES RNA in the cryo-EM reconstruction, the HLH motif in EIF3A is spatially distant from the EIF3D cap-binding domain [11] and is more discrete than the proposed multi-subunit interface thought to recognize specific RNA secondary structures [3] and $m^6$A modifications [5, 14]. We therefore probed the role of the EIF3A HLH motif in regulating cellular translation initiation in cells and *in vitro*. We found that mutations in the EIF3A HLH motif affected the translational efficiency of transcripts involved in proliferative pathways, including the mRNAs encoding MYC, PRL3 and MET. MYC is a well-known transcription factor strongly associated with cancer initiation and is found to be deregulated in over half of human cancers, whereas PRL3 and MET are implicated in cancer metastasis through regulating oncogenic effector pathways, such as PI3K/Akt/mTOR and HGF/SF signaling, respectively [15–17]. The selective enhancement of translation initiation on cancer-associated transcripts by the EIF3A HLH motif highlights a new mode of eIF3 translation regulation and identifies a well-defined, discrete structural motif [14] that could be targeted for future drug development efforts.

## Results

### Mutation of the EIF3A HLH motif causes selective translation repression of proliferative mRNAs

We first introduced the 3 amino-acid mutation in the HLH RNA-binding motif in EIF3A (KSKK > NSEE, hereafter termed EIF3A HLH*), previously shown to disrupt HCV IRES-

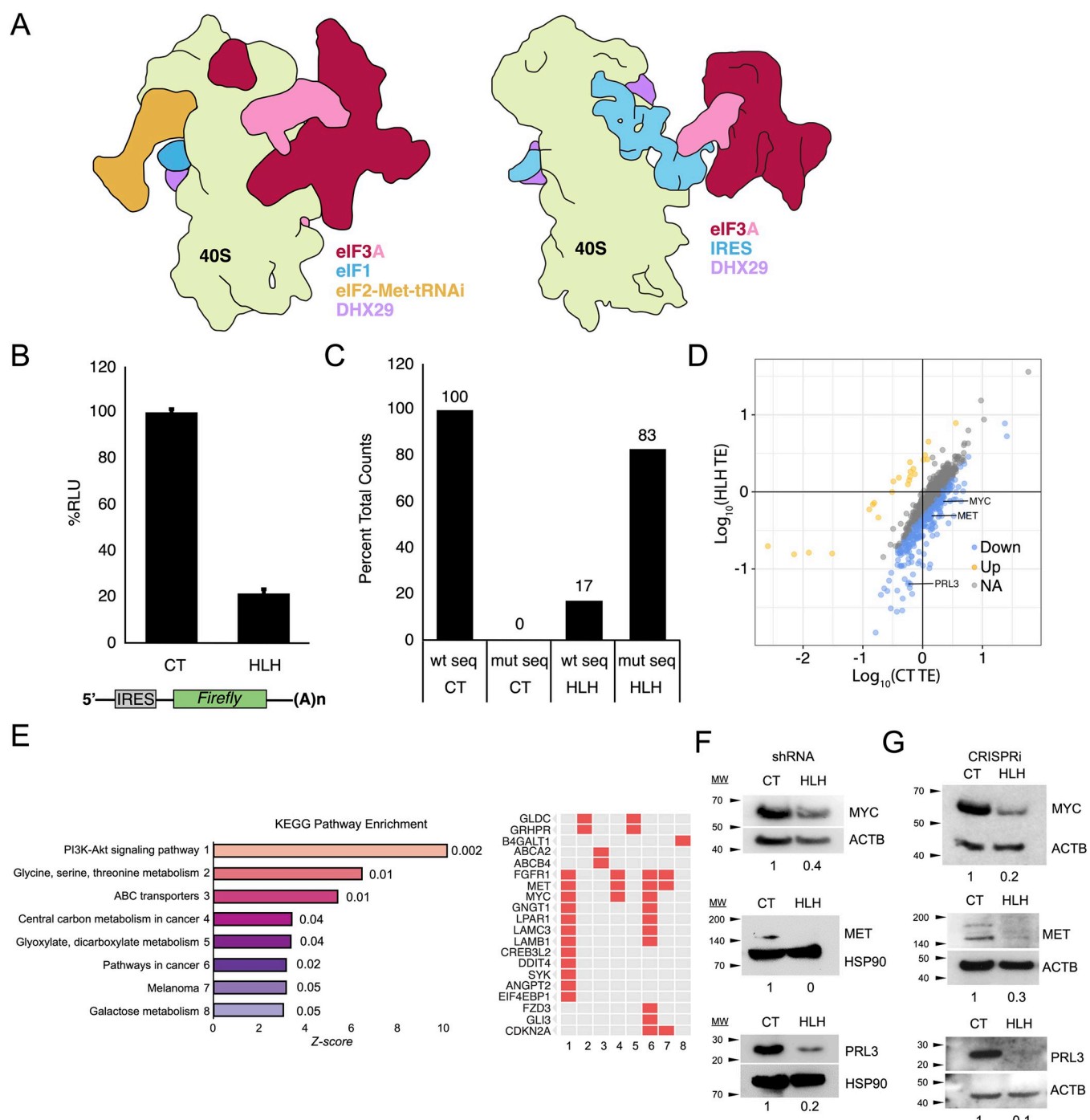

**Fig 1. EIF3A HLH motif regulates translation of proliferation-associated mRNAs.** A. Schematic of eIF3 binding to the 40S ribosomal subunit in canonical and viral IRES-mediated initiation complexes based on cryo-EM reconstructions [9, 13]. B. *In vitro* translation of IRES-*Renilla* mRNA in CT and HLH* extracts. Data represent mean ± S.D., n = 3 replicates. C. Representative alignment of RNAseq CT and HLH reads to wild type (wt) or HLH* (mut) *EIF3A* sequences. D. Translational efficiency scatter plot of statistically significant transcripts (p-value < 0.01 and >10 reads per transcript in the RNA-Seq analysis). Upregulated transcripts highlighted in gold (>2.5x increase), downregulated in blue (>2.5x decrease). E. KEGG Pathway enrichment analysis of downregulated transcripts, ranked by Enrichr generated Z-score (*p*-value listed next to bars) [52]. Genes corresponding to enriched categories are listed on the right. F. Representative western blot validation of top cancer associated hits in shRNA HEK293T cell lines. Protein quantification normalized to ACTB or HSP90 loading control and relative amounts indicated below corresponding bands. G. Representative Western blot in CRISPRi HEK293T cell lines. Protein quantification normalized to ACTB or HSP90 loading control and relative amounts indicated below corresponding bands. New MET antibody produced a double band not previously seen (see Experimental procedures).

mediated translation initiation [12], and tested for EIF3A HLH* incorporation into the endogenous eIF3 complex. We used transient expression of FLAG-tagged HLH* EIF3A and verified its incorporation, using an anti-EIF3B antibody pulldown of the eIF3 complex and Western blotting for EIF3A and the FLAG peptide (S1 Fig panel A in S1 File) [4]. For more detailed cellular analysis while avoiding potential assembly defects due to different expression levels of eIF3 subunits [18, 19], we introduced full length untagged EIF3A HLH* into HEK293T cells using a lentiviral delivery and integration system under hygromycin selection. We also generated control (CT) cell lines in the same manner but with no mutation in the exogenous EIF3A sequence. We then knocked down endogenous EIF3A expression using an shRNA targeting the native mRNA 3′ UTR, expressed by a second lentiviral system under dual hygromycin/ puromycin selection (Table 1, S1 Table in S2 File). HLH* cell lysates showed a dramatic (~80%) decrease in encephalomyocarditis virus (EMCV) IRES-mediated translation relative to the CT cell lysates, consistent with the effects previously seen *in vitro* with reconstituted eIF3 and the HCV IRES (Fig 1B) [12].

We then used the CT and HLH* cell lines to determine the effect of the EIF3A HLH mutation on the translational efficiency of cellular mRNAs. To assess the extent of mutant EIF3A expression, RNA deep sequencing (RNA-seq) data were aligned to wild-type and HLH* EIF3A sequences, revealing that >80% of aligned HLH reads mapped to the HLH* sequence, and confirming robust expression of exogenous over endogenous EIF3A (Fig 1C). HLH* EIF3A cells exhibited no change in global transcription, with only four genes exhibiting a significant change in expression (S1 Fig panels C-D in S1 File). However, translational efficiency analysis via ribosome profiling revealed a subset of transcripts differentially sensitized to HLH* EIF3A, 149 negatively regulated and 18 positively regulated (p-value < 0.01 and >2.5x change, Fig 1D, S2 Fig panel A in S1 File, S2-S4 Tables in S2 File). Functional classification of HLH-sensitive mRNAs showed significant enrichment for proliferative transcripts (Fig 1E, S2 Fig panel C in S1 File, S5 Table in S2 File), including those encoding PRL3, MYC and MET (Fig 1D). Consistent with the observed decreases in translational efficiency (Fig 1D, S2 Fig panel A in S1 File), western blot analysis showed a corresponding decrease in protein levels of MYC, MET, and PRL3 (Fig 1F, S2 Fig panel D in S1 File) in the mutant cells, suggesting that the HLH motif normally facilitates their translation.

When the mutant cells were passaged for longer timeframes, we observed that the shRNA cell lines expressing HLH* EIF3A developed resistance to the mutation and recovered MYC protein expression to levels similar to those seen in CT cells, with a consequent increase in their proliferative rate. We therefore engineered HEK293T with CRISPRi-mediated suppression of endogenous EIF3A expression, coupled with exogenous lentiviral CT or HLH* EIF3A expression, as in the shRNA cell lines [20] (Table 1, S1 Table in S2 File). In the CRISPRi cell lines, we were able to reproduce the decrease in MYC, MET, and PRL3 protein levels while improving cell line stability as assessed by growth rate and relative MYC levels as a function of passage number (Fig 1G, S2 Fig panel D in S1 File). The CRISPRi cell lines also exhibited a moderate global decrease in translation, as determined by metabolic labeling, indicating a stronger phenotype than the shRNA lines (S1 Fig panel B in S1 File). Therefore, we proceeded to use the CRISPRi cell lines for subsequent biochemical and luciferase reporter-based experiments.

## HLH* EIF3A causes transcript-specific defects in translation initiation factor recruitment

To assess the effect of HLH* EIF3A on translation initiation, we used the CRISPRi-edited CT and HLH* cell lines to prepare cytoplasmic extracts for *in vitro* translation experiments, using

**Table 1. Cell lines used in this study and the method of engineering.**

| Cell Line | Endogenous KD | Exogenous | Selection |
|---|---|---|---|
| CT | shRNA | eIF3A, shRNA | hygromycin, puromycin |
| HLH | shRNA | eIF3A HLH*, shRNA | hygromycin, puromycin |
| CT2SG3 | dCas9/sgRNA | dCas9, sgRNA, eIF3A | BFP, hygromycin, puromycin |
| HLH6SG3 | dCas9/sgRNA | dCas9, sgRNA, eIF3A HLH* | BFP, hygromycin, puromycin |
| R-CT | shRNA | eIF3A, shRNA | hygromycin, puromycin |
| R-HLH | shRNA | eIF3A HLH*, shRNA | hygromycin, puromycin |

the HLH-sensitive *MYC* mRNA compared to *GAPDH* as a control. We nuclease-treated the translation-competent extracts to degrade the endogenous RNAs and re-programmed the reactions with *in vitro* synthesized *GAPDH* and *MYC* mRNAs. We subsequently stalled the translation reactions with either cycloheximide or GMPPNP, and fractionated them on sucrose gradients (Fig 2A and 2B) [12]. Cycloheximide stalls 80S ribosomes immediately after initiation, whereas GMPPNP stalls 48S pre-initiation complexes at the start codon [12, 21]. We observed no change in initiation factor sedimentation in the CT lysates programmed with either *MYC* or *GAPDH* mRNAs by western blot analysis of the sucrose gradient fractions (Fig 2C–2F). However, we observed a defect in EIF3A and EIF5B distribution in the HLH* EIF3A *in vitro* translation reactions programmed with either *GAPDH* or *MYC* mRNA (Fig 2C–2F). These data are consistent with our previous results using reconstituted eIF3, which showed that the EIF3A HLH motif is important for eIF3 association with the 40S subunit, and for EIF5B association with pre-initiation complexes and the 80S ribosome when initiating translation with the HCV IRES [12], and suggest the EIF3A HLH motif plays an analogous role in translation initiation on certain cellular transcripts. We did not observe a defect in the distribution of Met-tRNA$_i$, in contrast to the previously observed defect *in vitro* using HCV IRES RNA with reconstituted eIF3 [12] (S3 Fig panels A-C in S1 File). Notably, *in vitro* translation reactions using HLH* EIF3A extracts programmed with *MYC* mRNA exhibited a specific defect in EIF4A1 incorporation into 48S pre-initiation complexes (Fig 2E and 2F), where EIF4A1 was observed to migrate at the top of the sucrose gradient rather than in the 40S fraction [4]. This result indicates that a defect in binding between the HLH motif and select mRNAs can propagate to a disruption of contact with EIF4A1, consistent with the previously observed coordination of roles of eIF3 and eIF4A1 for the translation of select transcripts in *Drosophila* [22]. The defect in relative EIF4A1 distribution occurred in both cycloheximide and GMPPNP stalled conditions, suggesting that the HLH mutation selectively destabilizes 48S pre-initiation complexes on a subset of mRNAs which are otherwise EIF4A1 dependent (Fig 2, S4 Fig panel A in S1 File) [4].

## HLH* EIF3A sensitizes translation extracts to EIF4A1 inhibitor RocA

To address the destabilizing effect of HLH* EIF3A on EIF4A1 incorporation into 48S pre-initiation complexes, we used Rocaglamide A (RocA) to inhibit EIF4A1 function in CT and HLH* extracts. RocA causes stalling, premature upstream initiation, and decreased translation by locking EIF4A1 onto poly-purine sequences in the 5′ UTR of mRNAs [23]. *In vitro* translation of *Renilla* luciferase reporter mRNAs harboring the HCV IRES or *MYC* 5′ UTR showed a marked translation defect in extracts from the HLH* cells (Fig 3A). By contrast, a *Firefly* luciferase reporter mRNA with the control beta-globin (*HBB*) 5′ UTR was unaffected by HLH* EIF3A, indicating that the HLH* mutation is specific to EIF3A HLH-sensitive mRNAs in a 5′ UTR dependent manner (Fig 3A). Notably, HLH* EIF3A sensitizes the HCV IRES and the

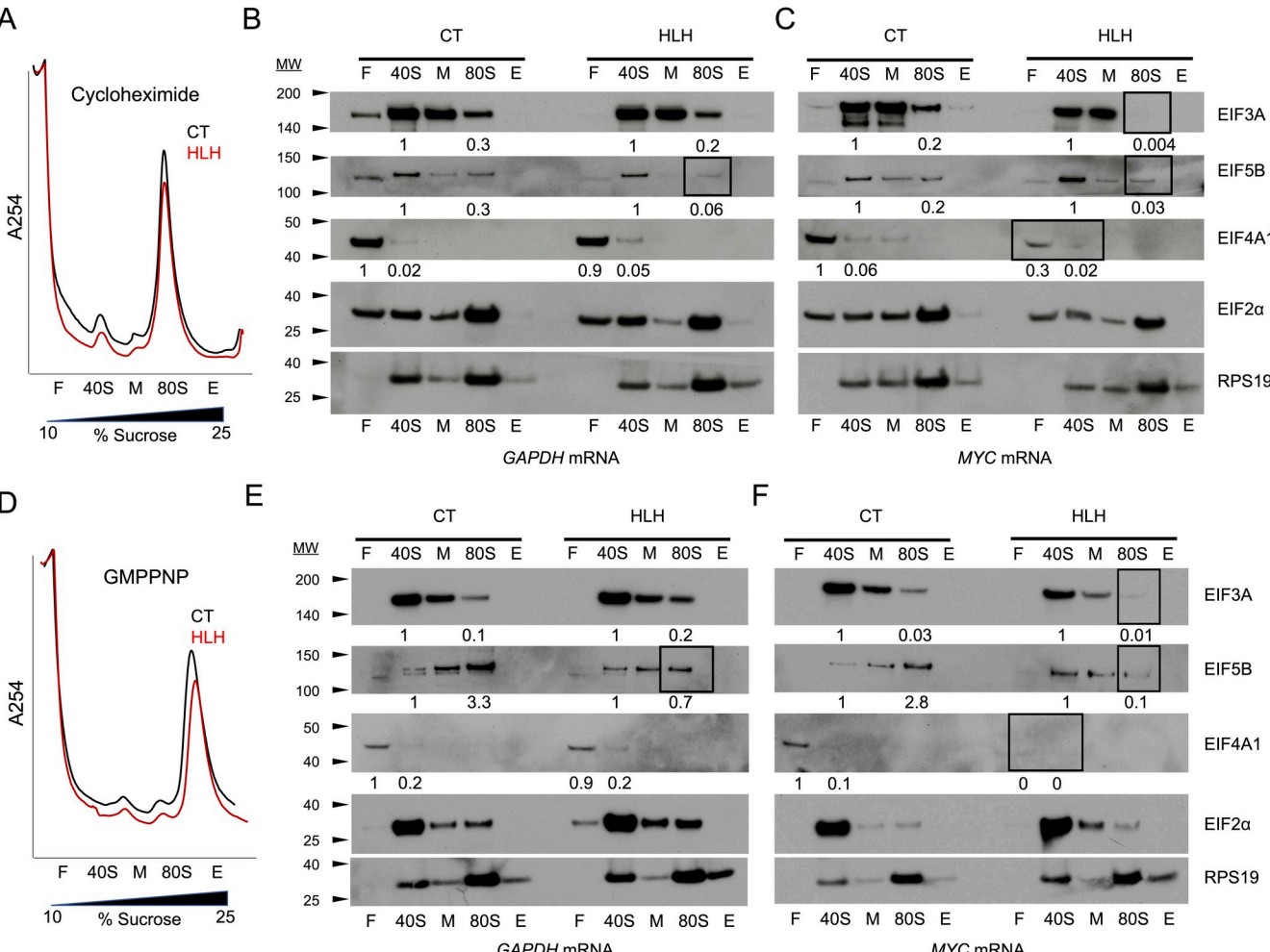

**Fig 2. EIF3A HLH mutation causes transcript-specific defects in initiation factor recruitment.** A, B. Representative sucrose gradient profile of *in vitro* translation reactions stalled with cycloheximide or GMPPNP and programmed with full length mRNA. Fractions were tracked by absorbance at 254 nm as shown, with the top of the gradient on the left. F: free RNA/top of gradient, 40S: 40S subunit, M: 60S/middle of gradient, 80S: 80S ribosome, E: end of gradient. Traces for exogenous WT EIF3A (CT, black) and for mutant HLH EIF3A (HLH, red) are shown. C, D. Representative western blot analysis of initiation factor distribution in translation reactions stalled with cycloheximide. Protein quantification normalized to RPS19 loading control and relative amounts indicated below corresponding bands. EIF4A1 was normalized to both RPS19 in the respective 40S fraction and to EIF2α, which gave equivalent ratios. E, F. Representative western blot analysis of initiation factor distribution in translation reactions stalled with GMPPNP. Protein quantification normalized as described in (C, D) and relative amounts indicated below corresponding bands. Boxes indicate fractions with decreased levels of initiation factors of interest.

*MYC* 5′ UTR to RocA in the *in vitro* translation reactions (Fig 3B). The RocA-dependent decrease in translation occurred in addition to the HLH*-specific defect (Fig 3A). Importantly, neither of these mRNAs were sensitive to RocA in the CT lysate, consistent with previous results [23], and consistent with HCV IRES not typically being dependent on EIF4A1 and thus insensitive to RocA [24].

To assess the effect of RocA on translation pre-initiation complex formation, we used *in vitro* translation reactions programmed with full-length *GAPDH* or *MYC* mRNAs and inhibited the reactions with both GMPPNP and RocA. Western blot analysis of the *in vitro* translation reactions fractionated on sucrose gradients showed that RocA treatment led to a dramatic decrease in EIF4A1 recruitment in the HLH* extracts re-programmed with *MYC* mRNA (Fig 3C–3E; compare to S4 Fig panels A and B in S1 File). Additional gradients to assess the

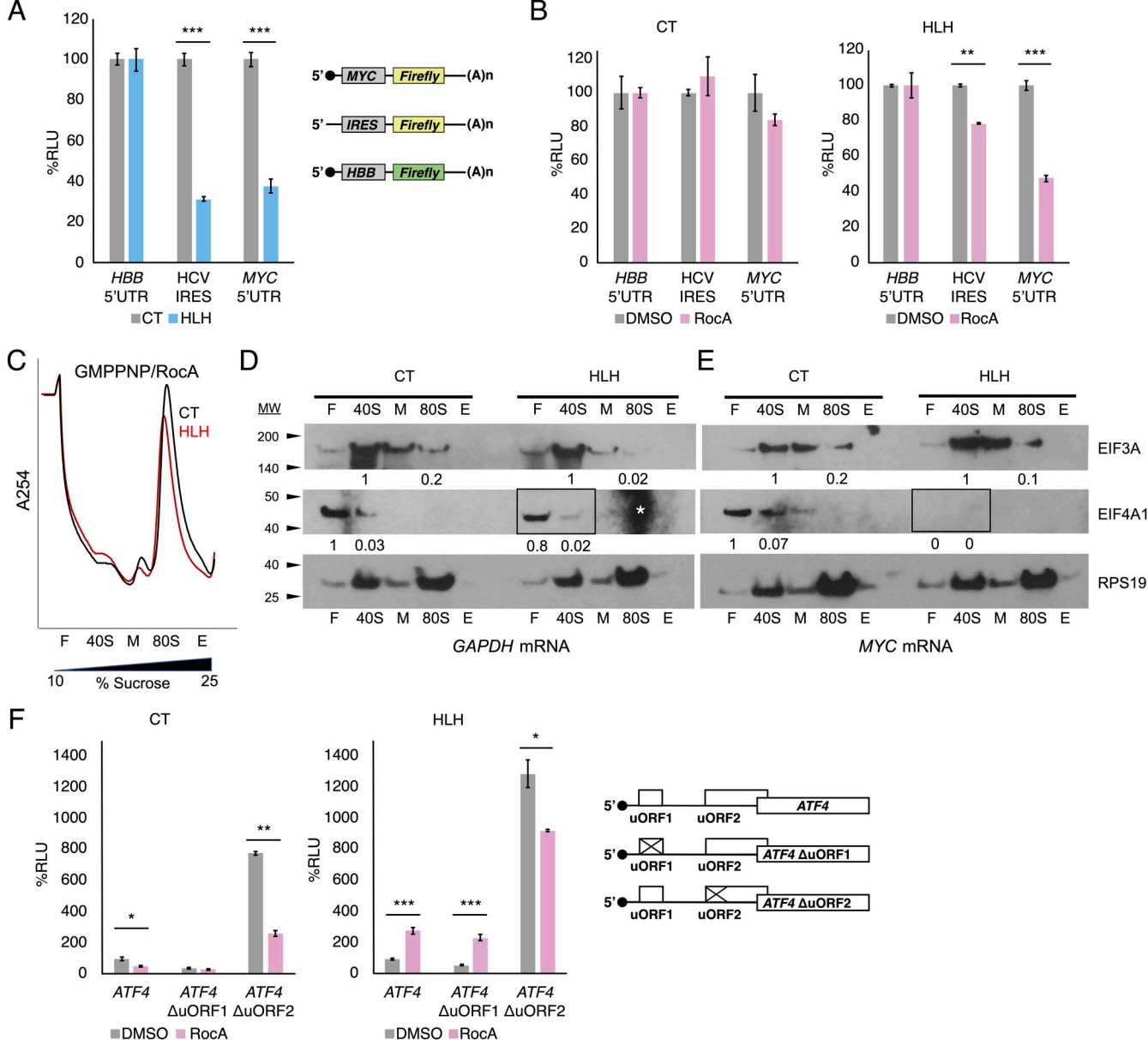

**Fig 3. EIF3A HLH motif selectively interacts with mRNA 5′ UTR elements.** A. *In vitro* translation of HCV IRES and *MYC* 5′ UTR *Renilla* mRNAs in CT and HLH* extracts. The schematic on the right shows the features and design of the mRNAs used. Globin *HBB* 5′ UTR *Firefly* mRNA was used as a control. ***P < 0.0001, Student's t-test. Data represent mean ± S.D., n = 3 replicates. B. *In vitro* translation of HCV IRES and *MYC* 5′ UTR luciferase mRNAs in the presence of EIF4A1 inhibitor RocA. *P < 0.01; **P < 0.001; ***P < 0.0001, Student's t-test. Data represent mean ± S.D., n = 3 replicates. C. *In vitro* translation reactions programmed with full length *GAPDH* and *MYC* mRNAs inhibited with both GMPPNP and RocA and fractionated on 10–25% sucrose gradients, as in Fig 2. Traces for exogenous WT EIF3A (CT, black) and for mutant HLH EIF3A (HLH, red) are shown. D and E. Western blotting of the sucrose gradient fractions, with boxes indicating fractions of interest for EIF4A1 levels. Protein quantification normalized to RPS19 loading control and relative amounts indicated below corresponding bands. Asterisk indicates background signal in gel, which does not interfere with initiation factor distribution analysis as there is no signal to obscure (see S4 Fig panel C in S1 File). F. EIF3A HLH* motif effects on RocA-mediated repression. The schematic on the right represents the *ATF4* uORF variant 5′ UTRs (WT, ΔuORF1, ΔuORF2) fused to the *Renilla* luciferase ORF for transfection into CRISPRi CT and HLH* cells. Live cell transfection was performed instead of *in vitro* translation in order to observe the effect of RocA stress. Relative Luciferase Units (RLU) percentage was normalized to internal globin *HBB* 5′ UTR Firefly mRNA control signal. *P<0.01, **P<0.001, ***P<0.0001, Student's t-test. Data represent mean ± S.D., n = 3 replicates.

migration of EIF4A1 in the sucrose gradients identified that much of EIF4A1 migrates at the top of the gradient (S4 Fig panel C in S1 File). The minimal effect of RocA on EIF4A1 distributions in *GAPDH*-programmed reactions or in CT extracts programmed with *MYC* mRNA (S4 Fig panels A and B in S1 File), suggests that the combined defect of the HLH* EIF3A and RocA on EIF4A1 association with 48S pre-initiation complexes is transcript-specific. No significant difference was observed for Met-tRNA$_i$ distribution, suggesting it is not perturbed by RocA (S4 Fig panel D in S1 File).

## EIF3A HLH motif interacts with mRNAs in counterpoint to EIF4A1

To address the apparent cooperation of the EIF3A HLH motif and EIF4A1 on select transcripts, we considered the functional role of the EIF3A HLH domain and EIF4A1 in relation to start codon recognition. In the yeast *Saccharomyces cerevisiae*, the EIF3A N-terminal domain (NTD), which includes the HLH motif, has been shown to enhance re-initiation upon translation of upstream open reading frames uORF1 and uORF2 of *GCN4* [25–27], likely by interacting with mRNA at the mRNA exit channel of the 40S subunit [28]. Notably, in addition to being tuned to levels of active eIF2, control of *GCN4* translation by the uORFs in its 5′ UTR also requires scanning in an eIF4A-dependent manner [29]. Translational control of the functional ortholog of Gcn4 in mammals–stress response transcription factor ATF4 –also requires uORF-mediated regulation [30]. Briefly, after translation of *ATF4* uORF1 under normal conditions, ribosomes re-initiate and continue scanning, encountering the inhibitory uORF2, which causes dissociation before the main ORF start codon [30]. Under stressed conditions with low levels of the eIF2 ternary complex, a portion of ribosomes scans through uORF2 and successfully initiates at the main ORF start codon. Deletion of uORF1 lowers translation under normal conditions, while deletion of uORF2 elevates it [30]. Unlike *MYC*, we did not identify *ATF4* as an HLH*-sensitive mRNA (Fig 1D, S2-S4 Tables in S2 File). However, we considered whether *ATF4* might become sensitized to HLH* EIF3A in the presence of RocA. We transfected CT and HLH* CRISPRi cells with *Renilla* luciferase reporter mRNAs harboring the *ATF4* 5′ UTR and saw a modest decrease in luciferase signal in the HLH* cells (S5 Fig panel A in S1 File). Upon treatment with RocA, translation increased in HLH* cells when compared to CT cells, for the mRNAs containing the WT *ATF4* 5′ UTR (WT) or the *ATF4* variant with a mutated start codon in uORF1 (ΔuORF1). RocA-mediated repression was also relieved in HLH* cells relative to CT cells for the *ATF4* variant with a mutated start codon in uORF2 (ΔuORF2) (Fig 3F). Treatment with thapsigargin (Tg), which induces ATF4 expression by decreasing active eIF2 levels [30] (S5 Fig in S1 File), also increased translation from the ΔuORF1 or ΔuORF2 *ATF4* reporter mRNAs in the presence of HLH* EIF3A (S5 Fig panels A and B in S1 File). By contrast, the combination of RocA and HLH* EIF3A exhibited a synergistic inhibitory effect on the translation of the *MYC* 5′ UTR in transfected cells (S5 Fig panel C in S1 File), consistent with what we observed in *in vitro* translation reactions (Fig 3B). Thus, while the HLH motif does not appear to regulate *ATF4* uORF translation on its own (Fig 3F), the mutant HLH* EIF3A can counterbalance EIF4A1 regulation of select transcripts, as reflected in the reversal of RocA sensitivity of the *ATF4* 5′ UTR.

## HLH* EIF3A suppresses MYC-induced proliferation of Burkitt's lymphoma cells

To investigate the impact of the HLH*-mediated defect in *MYC* mRNA translation, we selected a MYC driven cancer cell line in which to replicate the HLH* mutation and observe its impact on proliferation and cytotoxicity. We chose Ramos Burkitt's lymphoma cells [31], which are transformed due to MYC overexpression. We engineered Ramos Burkitt's

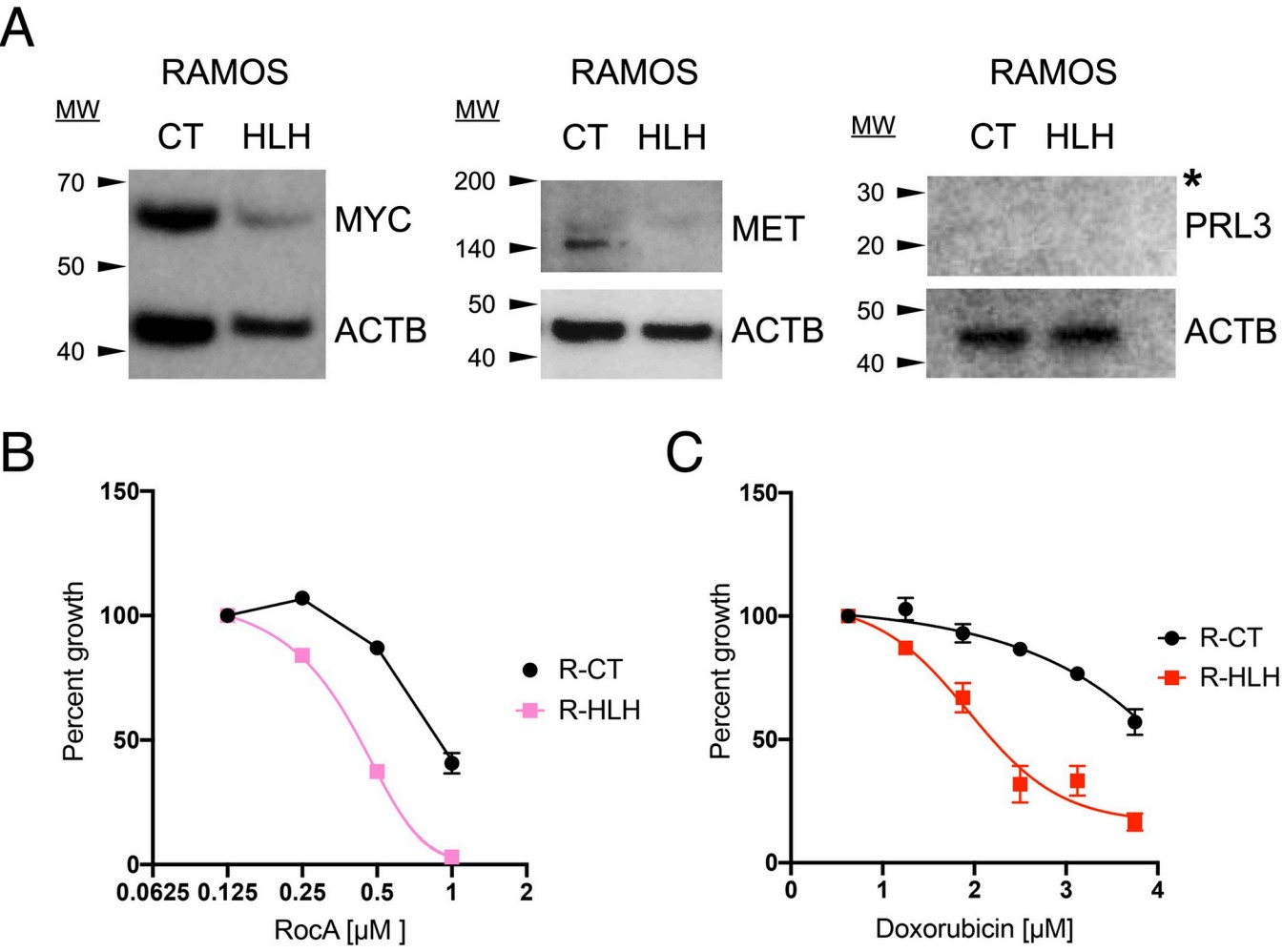

**Fig 4. EIF3A HLH mutation sensitizes Burkitt's lymphoma cells through loss of MYC.** A. Western blot analysis of HLH* sensitive proteins in Ramos shRNA lentiviral cell lines. Levels of protein normalized to ACTB control given below gels. Asterisk notes that no detectable PRL3 was found in Ramos cells. B. CT and HLH* Ramos cells cultured in the presence of increasing concentrations of RocA. Data represent mean ± S.D., n = 3 replicates. Some error bars are not visible due to being smaller than the data marker. C. Doxorubicin treatment of Ramos CT (R-CT) and HLH* (R-HLH) cell lines. Data represent mean ± S.D., n = 3 replicates. Some error bars are not visible due to being smaller than the data marker.

lymphoma cell lines expressing the CT or HLH* EIF3A using the shRNA method as in HEK293T cells and observed a substantial decrease in MYC and MET protein levels in the lymphoma cells expressing the HLH* EIF3A (Table 1, Fig 4A, S6 Fig panel A in S1 File). We also observed a global decrease in translation in the HLH* Ramos cell lines compared to the CT control, which was more pronounced than in the HEK293T cells (S6 Fig panel B in S1 File). To assess the combined effect of HLH* EIF3A and EIF4A1 inhibition, CT and HLH* Ramos cells were cultured overnight in the presence of increasing concentrations of RocA. The HLH* Ramos cells were highly sensitized to RocA compared to CT cells (Fig 4B). This is consistent with the synergistic sensitization of *MYC* 5′ UTR observed in *in vitro* translation extracts (Fig 3B), and may account for the *in vivo* effect due to Ramos cell dependence on MYC overexpression [31]. The sensitization effect of HLH* also occurred in the presence of chemotherapeutic agent doxorubicin, commonly used in clinical treatment of Burkitt's lymphoma (Fig 4C). Similarly to the HEK293T shRNA cells lines, in mutant Ramos cells passaged for longer timeframes (>2 weeks), we observed the cells expressing HLH* EIF3A upregulated

MYC protein expression, with a concomitant increase in their proliferative rate. To determine whether the exogenous expression of HLH* EIF3A was becoming suppressed, we performed total RNA-sequencing on CRISPRi HEK293T and shRNA Ramos cell lines (CT and HLH*, both) that had been passaged for >3 weeks. We were able to detect the presence of EIF3A mRNAs carrying the HLH mutation in both cell types. In CRISPRi HLH* cells, the relative abundance of wild type EIF3A remained low (~13% compared to HLH mutant reads), but in Ramos HLH* cells we observed upregulation of wild type EIF3A sequence that was nearly equal to CT cells (S6 Fig panel C in S1 File). These findings show that, within the course of a month, HLH* cells are able to develop resistance to dual antibiotic selection and restore the ability to proliferate through upregulation of wild type EIF3A, thus demonstrating the lethality of the HLH mutation.

## Discussion

Mammalian eIF3 has been shown to regulate translation initiation of specific mRNAs in a variety of ways: by binding RNA secondary structures in mRNA 5′ UTRs that activate or repress translation, through EIF3D binding to the m$^7$G cap, and through m$^6$A-dependent interactions with specific mRNAs [3–7]. Viral genomic RNAs also target eIF3 to promote translation initiation [32, 33]. Structural studies of HCV IRES binding and incorporation into translation preinitiation complexes revealed that the IRES displaces eIF3 from the 40S ribosomal subunit [13]. The HCV IRES-driven mode of interaction requires an HLH RNA binding motif in the EIF3A subunit of eIF3 that is critical for IRES binding and function. Mutation of the putative RNA-binding loop in the HLH motif of EIF3A disrupted eIF3 binding to the HCV IRES and to the 40S ribosomal subunit [12]. Here, using a combination of cell engineering and ribosome profiling, we show that mutating the loop in the HLH motif of EIF3A also negatively affects the translation of a discrete set of cellular mRNAs. The subset of cellular transcripts identified as functionally dependent on the EIF3A HLH motif does not overlap with the eIF3-dependent mRNAs identified previously that require RNA secondary structures [3], rely on EIF3D cap-binding [4, 5], or m$^6$A recognition [6, 7], suggesting that the HLH motif in EIF3A contributes to translation of these mRNAs using a unique mechanism.

In contrast to the HCV IRES, the cellular transcripts negatively affected by mutations in the HLH motif of EIF3A are not predicted to be enriched for putative viral-like RNA secondary structural elements in their 5′ UTRs to which eIF3 could bind [34], and are not enriched for uORFs [35, 36]. Although early studies suggested the MYC 5′ UTR exhibits IRES activity, this mechanism for MYC translation has since been definitively disproven [37, 38]. Since the mutation of the HLH motif in EIF3A also disrupts eIF3 binding to the 40S ribosomal subunit [12], translation of the cellular mRNAs identified here may be those most dependent upon the interaction of eIF3 with the 40S subunit. Structural and biochemical evidence has shown eIF3 interacts with the 40S subunit at both the mRNA entry and exit sites within pre-initiation complexes [28, 39, 40]. Specifically, the N-terminal domain of EIF3A binds the 40S subunit at the mRNA exit site, while the C-terminal domain projects towards the mRNA entry tunnel. In yeast, mutations in the N-terminal region of eIF3a that weaken mRNA binding to 48S pre-initiation complexes affect mRNA interactions at the mRNA entry channel, remote from where the eIF3a N-terminal region interacts with the 40S subunit. Conversely, mutations in the C-terminus of eIF3a that affect mRNA interactions with the mRNA entry channel also influence mRNA interactions with the mRNA exit site on the opposite side of the 40S subunit. These results reveal a long-distance connection between the two mRNA binding regions in the 48S pre-initiation complex important for mRNA recruitment [11]. The HLH motif in human EIF3A resides in the N-terminal region that binds at the mRNA exit site and, in yeast, is

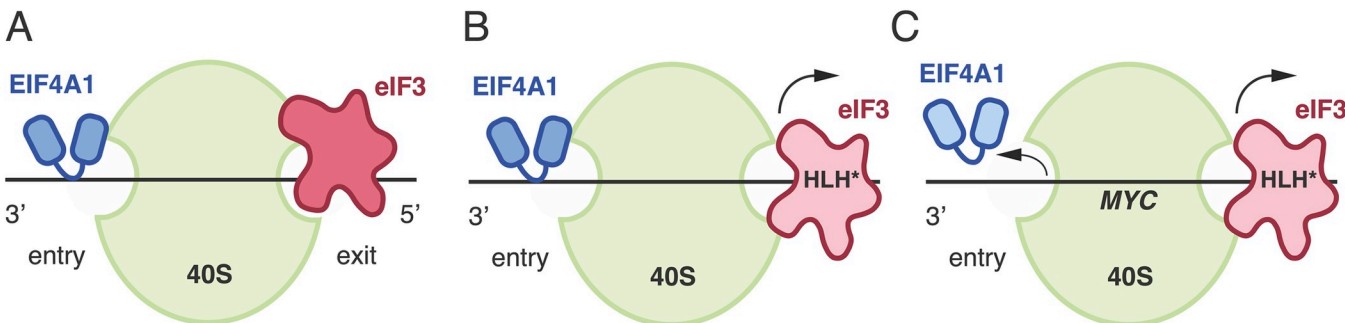

**Fig 5. Model of EIF3A dynamic interactions with EIF4A1 and variable mRNAs.** A. Schematic of EIF4A1 and eIF3 interacting with the mRNA at the entry and exit mRNA sites of the 40S ribosomal subunit, respectively. B. Schematic representing the displacement of EIF3 HLH* from the 40S subunit that leads to general defects in initiation factor recruitment and mRNA translation. C. Schematic representing the displacement of EIF3 HLH* from the 40S subunit in the presence of select transcripts, such as *MYC*, and the concomitant displacement of EIF4A1 at the entry site.

proposed to generally stabilize recruitment of mRNAs to the 43S pre-initiation complex [28]. By analogy to the yeast system, the disruption of eIF3 contacts with the 40S subunit by the 3-amino acid mutation in the HLH motif in human EIF3A [12] would be predicted to propagate to disrupt favorable interactions of PICs with cellular mRNAs in the mRNA exit channel (Fig 5A). Global elimination of eIF3-induced mRNA interactions at the mRNA exit site in complexes harboring HLH* EIF3A would then result in decreased translation initiation on transcripts with the least stable interactions in the mRNA entry channel (Fig 5A), possibly explaining the observed specificity in translation efficiency in HLH* EIF3A expressing cells. For the few mRNAs whose translation is upregulated in the presence of HLH* EIF3A (Fig 1E), additional experiments will be required to understand the interplay of the HLH motif in EIF3A and these mRNAs in the mRNA exit site.

In addition to uncovering cellular mRNAs that may be most sensitive to interactions with the 43S pre-initiation complex at the mRNA entry channel, we found that the HLH motif in EIF3A acts as a counterbalance to the action of EIF4A1 during translation initiation. eIF3 has been previously shown to interact selectively with eIF4A on subsets of mRNAs in a 5′ UTR specific manner [22] but the role of these interactions, as well as those of the eIF4F complex on these mRNAs, remains to be determined [41, 42]. We found that the HLH mutation destabilizes interactions of initiation complexes with EIF4A1 in a transcript-specific manner, possibly due to weakening eIF3 binding to pre-initiation complexes (Fig 2) [43]. Interestingly, recent structural evidence in mammals places the N-terminus of EIF3A at the mRNA exit site and EIF4A1 at the mRNA entry site [40]. The interaction of the HLH motif in EIF3A and EIF4A1 with the mRNA at the exit and entry points of the initiation complex, respectively, could affect the dynamics of mRNA scanning to the start codon in a transcript-specific manner, as well as affecting mRNA recruitment [24, 33]. This transcript specificity is enhanced in the presence of RocA, which locks EIF4A1 onto poly-purine stretches in the 5′ UTR in a dominant negative manner, thereby stalling scanning and mRNA unwinding [23, 44]. In the context of an HLH-sensitive transcript, RocA-mediated sequestration of EIF4A1 in non-productive eIF4F complexes [44] likely exacerbates the translational defect caused by HLH* EIF3A (Fig 3B) by further depleting EIF4A1 from the 48S PIC (Figs 2E, 2F, 3D and 3E, S4 Fig in S1 File). By contrast, for transcripts that are not highly reliant on the HLH motif in EIF3A, such as *ATF4*, RocA repression is alleviated by HLH* EIF3A. Taken together, these results suggest that loosening and tightening of mRNA contacts at either end of the mRNA channel in the 48S pre-initiation complex can lead to combinatorial increases or decreases in translation in a transcript-specific manner. The mRNA entry and exit channels are spatially separated [9, 11, 28],

implying that HLH* EIF3A loosening of mRNA contacts at the mRNA exit channel of the pre-initiation complex must propagate to the mRNA entry channel to affect the EIF4A1–mRNA interaction, connecting the functions of EIF3A and EIF4A1 during mRNA scanning (Fig 5). For example, the connection between the HLH domain in EIF3A and EIF4A1 could play a role in neuronal-specific translation, consistent with the coordinated role of eIF3 and eIF4A1 observed in *Drosophila* [22].

Several lines of evidence indicate that eIF3 dysregulation can contribute to cancer progression [45]. To test whether the molecular insights gained using HLH* EIF3A could have physiological implications in cancer, we generated HLH* EIF3A expressing Ramos Burkitt's lymphoma cell lines, which are dependent on MYC overexpression for their proliferation [31]. We observed that HLH* EIF3A resulted in a dramatic decrease in MYC protein levels, a global decrease in protein synthesis and a severe growth defect. Although a prior study suggested that indirect inhibition of MYC degradation by targeting glycogen synthase kinase 3β (GSK-3β) can increase doxorubicin sensitivity [46], others have shown that translation inhibition in chemoresistant lymphomas restores sensitivity to doxorubicin. These results were attributed to the loss of pro-survival proteins, such as c-MYC [47, 48], consistent with our findings of selective translation inhibition through defective EIF3A. The fact that HLH* EIF3A is sufficient to lower MYC levels in these lymphoma cells and increase their sensitivity to chemotherapeutic compounds suggests that eIF3 could serve as a potential target for cancer therapeutic strategies. However, further studies are required to interrogate the effect of HLH* EIF3A on MYC-addicted versus merely MYC-expressing cancer cells, as well as determine whether loss of MYC is the sole cause of the proliferation defect. Our model for how the HLH motif in EIF3A confers specificity on the translation of specific mRNAs involved in various pathways leading to cell proliferation makes it an intriguing target for treating a wide range of cancers. For example, by combining HLH* EIF3A with RocA treatment, we envision targeting both locations of mRNA engagement with the 48S pre-initiation complex, the mRNA entry and exit channels, to achieve selective and potent inhibition of translation. Furthermore, targeting the HLH motif in EIF3A could be used in conjunction with drugs that target parallel pathways, for example doxorubicin, a drug thought to inhibit transcription by targeting DNA topoisomerases [49], that is commonly used in the treatment of Burkitt's lymphoma. Taken together, the EIF3A HLH motif could be an attractive new target for drug development, both as a novel vulnerability of translationally active cancer cells as well as in combination therapy approaches.

## Experimental procedures

### Cell culture

HEK293T cells were cultured in DMEM (Invitrogen 11995–073) with 10% FBS (VWR Seradigm 97068–085) and Pen-Strep (10 U/mL) at 37˚C and 5% $CO_2$. HBSS was used for washing (Invitrogen 14175–103). Ramos cells were cultured in RPMI 1640 (Thermo 11875–119) with 10% FBS, 1 mM sodium pyruvate (Thermo 11360–070), 1x NEAA (Thermo 11140–050 100x), and 100 U/mL Pen-Strep (Thermo 15140–122). Lentiviral cell lines were selected with hygromycin (Thermo 10687010, 250 µg/mL) and puromycin (Mirus MIR5940, 10 µg/mL). CT or HLH* EIF3A was cloned into nLV103-hygro, and the custom EIF3A shRNA vector was obtained from pLKO.1-puro bacterial stock (see Supporting information). CRISPRi cell lines were made using catalytically dead Cas9 fused to BFP that was introduced into cells via lentivirus (see Supporting information). Transduced cells were FACS sorted into 96-well plates for clonal amplification, manually screened for BFP expression, and the brightest colonies selected for subsequent introduction of lentiviruses encoding sgRNA in pSLQ1371_BLP1_Ef1A_puro_GFP, a gift from the Jonathan Weissman Lab, and CT or HLH* EIF3A in nLV103-hygro

(see Supporting information). Lentiviral vectors were transfected into HEK293T cells to generate viral particles at a ratio of 1 μg: 250 ng: 250 ng lentivirus carrying the gene of interest, dR8.91 packaging vector, and pMD2. G envelope vector, respectively, per each well in a 6-well format. The viral supernatant was harvested at 48 hours and transduced into relevant cell lines using 20 μg/mL polybrene. DNA and RNA transfections were performed using Opti-MEM Reduced Serum Media (Invitrogen 31985–088) and TransIT-2020 (MIR5404) and TransIT-mRNA (MIR2225) reagents according to the manufacturer's protocol (Mirus). For all experiments, HLH* cell lines were compared to CT cell lines, to minimize artifacts that may arise from exogenous expression of eIF3 subunits [18, 19].

## Ribosome profiling

Ribosome profiling libraries were prepared from three biological replicates per cell line according to previously described methods [50]. RNAseq libraries were prepared from the same samples using TruSeq RNA Library Prep kit according to the manufacturer's instructions (Illumina Part # 15026495). All samples were analyzed for nucleotide length and concentration (Bioanalyzer). Raw Illumina reads from ribosome profiling and matched total RNA sequencing libraries were collapsed and adapters were trimmed using fastx_collapser from the FASTX Toolkit (Hannon, G.J. 2010. FASTX-Toolkit. http://hannonlab.cshl.edu/fastx_toolkit). Sequencing data were analyzed using Bowtie v1.0.0 [51] to remove rRNA reads, TopHat v2.0.14 [52, 53] to align reads to the human GRCh38 genome, Cufflinks v2.2.1 and Cuffdiff v2.2.1 [52] to extract and merge raw read counts of the biological replicates, and R v3.2.2 package Babel v0.2–6: Ribosome Profiling Analysis to calculate FPKM, *p*-values and FDR (R Core Team (2015). R: A language and environment for statistical computing. R Foundation for Statistical Computing, Vienna, Austria https://www.R-project.org/). Translational efficiency (TE) was calculated as ribosome profiling FPKM / RNAseq FPKM and fold change was calculated as HLH* TE / CT TE. See S2-S4 Tables in S2 File for data (raw non-zero, Babel output). Bowtie2 [54] was used to generate indices from EIF3A CT and HLH* sequences for alignment with CT and HLH* RNAseq data (see Supporting information). All data has been deposited in Gene Expression Ombinus (accession number GSE118239). Functional classification was carried out by hand and is included in S5 and S6 Tables in S2 File. KEGG Pathway enrichment was performed using Enrichr (http://amp.pharm.mssm.edu/Enrichr/) [55]).

## RNA sequencing

RNAseq libraries were prepared from the biological duplicates per cell line (CRISPRi CT and HLH*, Ramos CT and HLH*) using TruSeq RNA Library Prep kit according to the manufacturer's instructions (Illumina Part # 15026495). All samples were analyzed for nucleotide length and concentration (Bioanalyzer). Raw Illumina reads were collapsed and adapters were trimmed using fastx_collapser from the FASTX Toolkit. Raw sequencing data was analyzed manually to identify reads corresponding to CT or HLH* EIF3A target sequence (see Supporting information).

## 5′ UTR uORF and secondary structure computational analysis

We used the databases uORFdb [35] (uORFdb—a comprehensive literature database on eukaryotic uORF biology, http://www.compgen.uni-muenster.de/tools/uorfdb/) and TISdb [36] (Translation Initiation Site Database, http://tisdb.human.cornell.edu/) to analyze transcripts affected by HLH* EIF3A for uORF presence. We used RNAstructure v6.0.1 Secondary Structure Web Server [34] (https://rna.urmc.rochester.edu/RNAstructureWeb/) to predict secondary structures of transcripts affected by HLH* EIF3A.

## Western blotting

The following antibodies were used for Western blot analysis using the manufacturers' suggested dilutions: anti-beta-Actin (Abcam ab8227), anti-ATF4 (Abcam ab184909), anti-DEP-TOR (Sigma SAB4200214), anti-eIF1A (Abcam ab177939), anti-eIF2α (Bethyl A300-721A-M), anti-eIF3A (Sigma SAB1402997-100UG), anti-eIF4A1 (Abcam ab31217), anti-eIF5B (Bethyl A301-745A-M), anti-FLAG M2-Peroxidase (HRP) (sigma A8592), anti-HSP90 (Abcam ab13492), anti-MET (EMD Millipore 07–283, Abcam ab51067), anti-MYC (Abcam ab32072), anti-PTP4A3 (Abcam ab50276 recognizes PRL-3), anti-RPS19 (Bethyl A304-002A), anti-Mouse IgG-HRP (A00160), anti-Rabbit IgG-HRP (NA934V). Protein levels in Western blots were quantified using ImageJ [56].

## EIF3B immunoprecipitation

Immunoprecipitation of eIF3B was performed according to previously published methods [3, 57]. Briefly, HEK293T cells were transfected with CT or HLH* EIF3A in nLV103-hygro (see Supporting information) in biological triplicates, passaged for 24, 48, and 72 hours, and collected at ~80% final confluency for lysate preparation. Cells were washed and scraped in cold PBS, spun down for 5 min at 1000 g at 4˚C, and resuspended in 3x (relative to pellet weight) NP40 lysis buffer (50 mM HEPES pH 7.5, 150 mM KCL, 2 mM EDTA, 0.5 mM DTT, 0.5% NP40). Lysates were incubated on ice for 10 min, spun down for 10 min at 14,000 g at 4˚C, and the supernatant removed, avoiding the pellet. A quarter of the lysate volume was reserved as input and the remainder was brought up to 400 μL with NP40 lysis buffer. Dynabeads Protein G (Thermo 10003D) and anti-eIF3B antibody (Bethyl A301-761A) were used for precipitation of EIF3 complexes. 25 μL of Dynabeads was prepared for each 400 μL lysate sample by washing 2x with 1x PBS + 0.02% Tween-20 on a magnetic rack. 5 μL of antibody in 150 μL 1x PBS + 0.02% Tween-20 was added to beads and allowed to bind while rotating for 30 min at RT. Bound beads were washed 2x with NP40 lysis buffer on a magnetic rack, resuspended with 400 μL of lysate sample, and incubated while rotating for 2 hrs at 4˚C. Bound samples were washed 4x with 350 μL wash buffer (50 mM HEPES pH 7.5, 250 mM KCL, 0.5 mM DTT, 0.5% NP40) and resuspended in NuPAGE LDS Sample Buffer (Thermo NP0007) according to manufacturer's instructions. Samples were boiled for 5 min at 95˚C, resolved on SDS-PAGE gels and analyzed by Western blotting.

## *In vitro* transcription

RNAs were transcribed from 1 μg of PCR-amplified templates using T7 RNA polymerase in 1x transcription buffer (30 mM Tris-Cl pH 8, 5 mM MgCl$_2$, 0.01% Triton X-100, 2 mM spermidine, 20 mM NTPs, 10 mM DTT) for 5 hrs at 37˚C. Reactions were treated with RQ1 DNAse (Promega M6101) for 20 min at 37˚C, precipitated using 2x volume 7.5 M LiCl/50 mM EDTA at -20˚C for 30 min, washed 2x in 70% EtOH, and resuspended in RNase free water. RNAs were capped using the Vaccinia capping system (NEB M2080S) according to manufacturer's protocol, in the presence of 100 U murine RNase inhibitor (M0314S), extracted with an equal volume of phenol:chloroform pH 6, precipitated at -20˚C overnight in 5x volume 2% LiClO$_4$ in acetone, washed 2x in 70% EtOH and resuspended in RNase free water. RNAs that were amplified without a poly-A tail were poly-adenylated using poly-A polymerase (NEB M0276) according to the manufacturer's protocol.

## *In vitro* translation

Cell extracts were prepared from CRISPRi-engineered HEK293T cell lines at ~80% confluency. Cells were washed and scraped in cold PBS, spun down for 5 min at 1000 g at 4˚C,

and resuspended in an equal volume of hypotonic lysis buffer (10 mM HEPES 7.6, 0.5 mM MgOAc, 5 mM DTT, Halt protease/phosphatase inhibitor cocktail (Thermo 78440)) for 45 min. Extracts were homogenized ~20 times through a 27G needle, spun down for 1 min at 14,000 g at 4˚C, and the supernatant removed, avoiding the lipids on the top and interface on the bottom. *In vitro* translation reactions with luciferase reporter mRNAs were carried out with 0.5x extract, energy mix (final 0.84 mM ATP, 0.21 mM GTP, 21 mM creatine phosphate (Roche 10621722001), 45 U/mL creatine phosphokinase (Roche 10127566001), 10 mM HEPES pH 7.5, 2 mM DTT, 2.5 mM MgOAc, 50 mM KOAc, 8 μM amino acids (Promega PRL4461), 255 μM spermidine, 1U/mL murine RNase inhibitor (NEB M0314)), and 400 ng total RNA. Rocaglamide A (RocA, a gift from the Nicholas Ingolia Lab) was added to a final concentration of 0.1 μM where indicated. Reactions were incubated for 1 hr at 30˚C and luciferase signal was measured using Dual-Glo Luciferase Assay System (Promega E2920).

*In vitro* translation extracts for sucrose gradient fractionation were first treated with micrococcal nuclease (NEB M0247S) and 0.8 mM $CaCl_2$ for 10 min at 25˚C. Treatment was stopped with 3.2 mM EGTA. Treated extracts were then mixed in a 2:1:1 ratio with the energy mix and 1 μg full-length *MYC* or *GAPDH* mRNA in water and incubated for 20 min at 30˚C prior to loading on gradients. Cycloheximide (100 μg/mL; Sigma 01810) or GMPPNP (Sigma G0635) were added to the energy mix prior to the translation reaction. GMPPNP was added at 0.21 mM instead of GTP.

## Sucrose gradient fractionation

*In vitro* translation reactions using extracts prepared from CRISPRi-engineered cell lines were sedimented on 10–25% sucrose gradients (containing 20 mM HEPES pH 7.5, 150 mM KOAc, 2.5 mM MgOAc, 1 mM DTT, 0.2 mM spermidine, 100 μg/mL cycloheximide if reaction contained cycloheximide) for 3.5 hrs at 240,000 g at 4˚C using a SW41 rotor (Beckman Coulter). Gradients were fractionated using Teledyne Isco Tris Peristaltic Pump and fractions were collected and pooled according to the UV trace. Fractions were concentrated using Amicon 30 kDa spin columns (UFC503096) according to the manufacturer's instructions. For western blot analysis, fractions were quantified for total protein concentration and equal amounts were loaded per well. For Northern blot analysis, fractions were treated with 1% SDS and 1% Proteinase K solution (20 mg/mL Proteinase K (Thermo 26160), 0.2 M Tris-HCl pH 7.5, 0.2 M NaCl, 1.5 mM $MgCl_2$) at 42˚C for 30 min. RNA was extracted using an equal volume of phenol:chloroform pH 6, precipitated at -20˚C overnight in 2x volume 100% EtOH, 2.7 M NaOAc, and 10 μg/mL GlycoBlue Coprecipitant (Thermo AM9515), washed 2x in 70% EtOH and resuspended in RNase free water.

## Northern blotting

Total RNA isolated from the sucrose gradient fractions was resolved using a 10% polyacrylamide gel in 0.5x TBE buffer buffer (1x TBE buffer contains 89 mM Tris, 89 mM boric acid, and 2 mM EDTA) and electroblotted onto a nylon (N+) membrane (GE Healthcare RPN203B) at 20 V for 90 min at 4˚C in 0.5x TBE buffer. The membrane was crosslinked and pre-hybridized in UltraHyb Hybridization Solution (Thermo AM8670) at 42˚C for 1 hour, then incubated overnight with 50 pmol Met-tRNA$_i$ specific probe (5′–TGGTAGCAGAGGATGGTTTCGAT–3′). The probe was labeled on the 5′ end with [γ- 32P] ATP (Perkin Elmer) using T4 polynucleotide kinase (NEB M0201) according to the manufacturer's protocol. Membranes were washed twice by 20 mL 6x SSC for 5 min at 42˚C and twice by 20 mL 2x SSC and twice by 20 mL 1x SSC (20x SSC contains 0.3 M sodium citrate in 3 M NaCl). Membranes were then wrapped in saran wrap, exposed to a phosphor screen overnight, and visualized by phoshor-imaging.

## Cell viability assays

Ramos cells were seeded at 1 x $10^6$ cells/mL into 96-well plates in the presence or absence of drug (RocA, gift from Nicholas Ingolia Lab, 0–0.1 μM; Doxorubicin (Fisher BP25161), 0–4 μM), cultured for 24 hours, and cell viability was assessed using CellTiter-Glo assay according to the manufacturer's protocol (Promega G7570).

## Metabolic labeling

Cells were seeded at 1 x $10^6$ cells/mL into 6-well plates and allowed to adhere and grow overnight. Media was changed to DMEM–Met/Cys (Thermo 21013024) for 30 min, then each well was incubated for 30 min at 37˚C with 5 μl/well EXPRE35S35S Protein Labeling Mix (PerkinElmer NEG072002MC), after which cells were lysed in RIPA lysis buffer (50 mM Tris-HCl pH 7.5, 150 mM NaCl, 0.25% deoxycholic acid, 1% NP-40, 1 mM EDTA). Lysates were boiled in SDS loading buffer, resolved on SDS-PAGE gels, and stained with Coommassie to visualize total protein. Gels were dried at 80˚C for 1 hour on a gel drier, exposed to a phosphor screen overnight, and $^{35}$S incorporation was visualized by phoshor-imaging.

**Plasmids and gene sequences** are included in Supporting information.

## Supporting information

**S1 File.**
(PDF)

**S2 File.**
(XLSX)

**S1 Raw images.**
(PDF)

## Acknowledgments

The authors thank Amy S.Y. Lee, Nathanael Lintner, M. Duane Smith, Audrone Lapinaite and Michael Rape for helpful discussions. We thank Mia Pulos-Holmes for help analyzing cell line stability. We also thank Jonathan Weissman and Jacob Corn for CRISPRi-related plasmids, and Nicholas Ingolia and Shintaro Iwasaki for RocA and helpful feedback.

## Author Contributions

**Conceptualization:** Marina P. Volegova, Jamie H. D. Cate.

**Data curation:** Marina P. Volegova.

**Formal analysis:** Marina P. Volegova, Jamie H. D. Cate.

**Funding acquisition:** Jamie H. D. Cate.

**Investigation:** Marina P. Volegova, Cynthia Hermosillo, Jamie H. D. Cate.

**Methodology:** Marina P. Volegova, Jamie H. D. Cate.

**Project administration:** Jamie H. D. Cate.

**Resources:** Jamie H. D. Cate.

**Supervision:** Jamie H. D. Cate.

**Validation:** Marina P. Volegova, Cynthia Hermosillo.

**Visualization:** Marina P. Volegova, Jamie H. D. Cate.

**Writing – original draft:** Marina P. Volegova, Jamie H. D. Cate.

**Writing – review & editing:** Marina P. Volegova, Cynthia Hermosillo, Jamie H. D. Cate.

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
