## [Decision Letter · Decision Letter 0]

16 May 2023

PONE-D-23-10825The Helix-Loop-Helix motif of human EIF3A regulates translation of proliferative cellular mRNAsPLOS ONE

Dear Dr. Cate,

Thank you for submitting your manuscript to PLOS ONE. After careful consideration, we feel that it has merit but does not fully meet PLOS ONE’s publication criteria as it currently stands. Therefore, we invite you to submit a revised version of the manuscript that addresses the points raised during the review process.

Your manuscript was reviewed by three experts in the field. The reviewers were in general positive, however, a number of concerns were brought up. The concerns warrant merit and addressing these issues with further experiments would help support the conclusions. Specifically, controls were lacking (i.e. western blots, sucrose gradients, # of replicates) and the rationale for experiments should be clarified more (why specific transcripts were chosen). Other concerns include the status of the in vitro translation extract that should be addressed. I believe that addressing all of the reviewers' comments through experiments and/or responses would strengthen the manuscript.  Also, please go through carefully the labeling of the figures in the figure legend.  Please ensure that your decision is justified on PLOS ONE’s publication criteria and not, for example, on novelty or perceived impact.

We look forward to receiving your revised manuscript.

Kind regards,

Eric Jan, Ph.D.

Academic Editor

PLOS ONE

Journal Requirements:

Reviewers' comments:

Reviewer's Responses to Questions

**Comments to the Author**

1. Is the manuscript technically sound, and do the data support the conclusions?

Reviewer #1: Partly

Reviewer #2: Yes

Reviewer #3: Yes

2. Has the statistical analysis been performed appropriately and rigorously? 

Reviewer #1: No

Reviewer #2: No

Reviewer #3: Yes

3. Have the authors made all data underlying the findings in their manuscript fully available?

Reviewer #1: Yes

Reviewer #2: Yes

Reviewer #3: No

4. Is the manuscript presented in an intelligible fashion and written in standard English?

Reviewer #1: Yes

Reviewer #2: Yes

Reviewer #3: Yes

5. Review Comments to the Author

Reviewer #1: In this manuscript, the authors report their studies on the helix-loop-helix motif of eIF3a n regulating eIF3a function in translational control of cancer-relevant mRNAs such as Myc. While some findings are interesting and worthy pursuing, it appears that this manuscript was put together with data that do not show cohesiveness and logic. It also lacks rigor for publication.

1. The authors changed silencing methods from shRNA to CRISPRi. However, RNAseq to verify knockdown of endogenous eIF3a by CRISPRi is missing.

2. Although Figure S1A shows a pulldown of Flag-eIF3a with eIF3b, the pulldown (co-IP) to show interaction between untagged eIF3a and eIF3b, which were used for all subsequent studies is missing.

3. The fractionation and Western blot appear to have been done only once with most western blot shown in low quality including unexplained asterisks on the blot. As more scientific rigor is required by NIH, this quality of work supported by NIH does not meet its standards.

4. The rationale in studying the selected eIFs (Fig. 2) is not clearly stated.

5. Sensitization of by RocA is confusing since HLH*eIF3a already released eIF4A1. This study needs discussion and explanation.

6. Study of ATF4 is also confusing. Firstly, it is not one of the genes found in the sequencing data. Secondly, it is not relevant to the study of Myc regulation or Myc function in the subsequent studies on cancer. Perhaps, similar studies should be performed for Myc but it is unclear if Myc mRNA has uORFs.

7. While the finding that HLH*eIF3a makes RAMO cells proliferate slower and more sensitive to doxorubicin is interesting, this study barely scratched the surface. It is unclear how reduction of Myc by HLH*eIF3a sensitizes RAMO cells to doxorubicin. Slow growing cells are normally more resistant to drugs such as doxorubicin. The author’s finding is also inconsistent with literature where Myc stabilization or upregulation sensitizes cells including RAMO to doxorubicin.

Reviewer #2: The manuscript titled “The Helix-Loop-Helix” motif of human EIF3A regulates translation of proliferative cellular mRNAs” is very well written, and thoughtful control experiments are performed. For example, authors used a lentiviral vector to express EIF3F HLH* under hygromycin selection and then depleted endogenous EIF3A using 3’UTR-specific shRNA under dual hygromycin/puromycin selection. Another good example is when the authors used CRISPRi-mediated depletion of endogenous eIF3A and then expressed HLH* using a lentiviral vector. Significant and meaningful ribosome profiling data has been generated which serves as the basis for selecting three different mRNAs to look at. Overall, this is a very nice mechanistic manuscript. However, I have some comments and questions as listed below. Some of these do require further experimentation.

For Fig. 1C: The authors should perform western blot analysis to show the depletion of endogenous eIF3A and expression of FLAG HLH*.

The authors have checked the IRES activity in Fig 1B using EMCV. They should also check the IRES activity of myc 5’ UTR, as it is very well known that myc expression is regulated by an IRES element.

Authors showed in their ribosome profiling experiment that they found 149 negatively regulated and 18 positively regulated mRNAs. However, they have not provided a clear rationale as to why they chose three mRNAs (out of 167 mRNAs) to follow up on.

In their CRISPRi-mediated depletion of endogenous eIF3A and subsequent exogenous HLH* expression using lentiviral vector experiment the authors should show the level of eIF3A (endogenous) and HLH* by western blotting (PDF page # 12).

After nuclease treatment of their cell lysates, the authors need to confirm that the nuclease activity has been effectively terminated by EGTA treatment. I see that they have used EGTA but then did not confirm that the nuclease activity has been stopped.

In the ribosome fractionation experiments, the authors have not shown the presence of GAPDH or MYC mRNAs presence in any of their fractions (48S and 80S). This should be done otherwise it becomes difficult to imagine that ribosome is actually recruited on these mRNA, particularly when the authors did not show the effectiveness of EGTA treatment of their lysates.

In the same figure (Fig 2) the authors should perform fractionation of their control lysate WITHOUT priming with GAPDH or MYC mRNAs.

Are both GAPDH and MYC mRNA capped? If so, the authors should perform this experiment with A-capped myc mRNA.

On PDF page # 12, (manuscript page 6), the authors claim that they have observed a ‘defect’ in eIF3A and eIF5B distribution …. On what basis? Amount of these proteins detected by western blots? Band density? If so, they have not normalized them to loading control.

The authors did not show the interaction of eIF3A, and the lack of HLH*, with MYC mRNA. They should show this by performing REMSA or filter binding assays.

The authors have used only one Ramos Burkitt’s Lymphoma cell line, but what if the effect they are seeing is cell line specific? They need to perform the cell viability assays in at least three cell lines.

Other Figure-specific comments:

They should use the standard error of mean and t-test in all their biological replications and not the standard deviation of biological triplicates.

The molecular wt mentioned for eIF5B in Fig 2 seems to be inaccurate.

uORF2 should overlap with the CDS for ATF4 gene organization figure (Fig 3F). I have checked their mRNA sequence in the supplementary file, which seems to be correct. However, the presentation of the ATF4 organization in Fig 3F is incorrect.

Important: in Fig 4 the authors claim that HLH* mutation sensitizes Burkitt’s lymphoma cells through loss of MYC. However, they have not established a clear link for the effect on cell viability. Yes, they do see significant cell death in HLH* with doxorubicin treatment. However, this cannot be only attributed to the loss of MYC. In order to claim that the authors should perform an experiment where they exogenously express MYC in HLH* mutant background and perform the same visibility assay. The reversal of cell death in the suggested condition would suggest the sensitization to doxorubicin was actually through MYC.

Minor points:

PDF page # 9, it should be in contrast with not “By contrast with…”

Fig S1: did they perform transient transfection of lentivirus or transduced lentivirus?

Reviewer #3: The manuscript by Volegova et al. investigated the role of the Helix-Loop-Helix (HLH) RNA binding motif of EIF3A in regulating translation of cellular mRNAs. The HLH motif was previously known to aid IRES-mediated translation of Hepatitis C Virus. In this study, the authors reported that the HLH motif regulates the translation of a subset of oncogenic mRNAs, notably MYC. Using HLH mutant cells, the authors showed that the translation of MYC, MET and PRL3 is reduced. Mechanically, they proposed that the HLH motif modulates the interaction between eIF3A and EIF5B, and also affects the interaction between EIF4A1 and 43S pre-initiation complex (PIC).

By focusing on the role for the HLH motif of EIF3A, it is interesting to find that translation of a small subset of genes is affected by the EIF3A mutant. The proposed mechanism differs from IRES, although it’s unclear how the HLH domain of EIF3A modulates the helicase EIF4A1 in the scanning process. Overall, the manuscript was well-written and the results were nicely presented. The manuscript would be improved if the following concerns are addressed.

Major concerns:

1. Figure 2, it would be nice to show the sucrose gradient profiles in the presence of wild type or HLH mutants. This is important to demonstrate the overall translation capacity.

2. Similarly, what does the polysome profile look like in CRISPRi-edited cell lines?

3. Figure 4, the protein level of ACTB are greatly reduced in HLH RAMOS cells. Is the presence of HLH mutant toxic to the cell? Besides cancer cells, is the presence of HLH mutant toxic to normal cells as well?

6. PLOS authors have the option to publish the peer review history of their article (what does this mean?). If published, this will include your full peer review and any attached files.

Reviewer #1: No

Reviewer #2: No

Reviewer #3: No

---

## [Author Response · Author response to Decision Letter 0]

19 Jul 2023

Please see file included with the manuscript files.

---

## [Decision Letter · Decision Letter 1]

1 Sep 2023

PONE-D-23-10825R1The Helix-Loop-Helix motif of human EIF3A regulates translation of proliferative cellular mRNAsPLOS ONE

Dear Dr. Cate,

Thank you for submitting your manuscript to PLOS ONE. After careful consideration, we feel that it has merit but does not fully meet PLOS ONE’s publication criteria as it currently stands. Therefore, we invite you to submit a revised version of the manuscript that addresses the points raised during the review process.

Your manuscript was reviewed by two reviewers in the field. In principal, the reviewers and I are in agreement that all comments and concerns are addressed. However, one reviewer suggests that the authors edit the discussion to take into account of different mechanism/interpretation pertaining to the RAMO cell response to doxorubicin. I welcome you to provide a brief minor discussion on this.

We look forward to receiving your revised manuscript.

Kind regards,

Eric Jan, Ph.D.

Academic Editor

PLOS ONE

Journal Requirements:

Reviewers' comments:

Reviewer's Responses to Questions

**Comments to the Author**

1. If the authors have adequately addressed your comments raised in a previous round of review and you feel that this manuscript is now acceptable for publication, you may indicate that here to bypass the “Comments to the Author” section, enter your conflict of interest statement in the “Confidential to Editor” section, and submit your "Accept" recommendation.

Reviewer #1: All comments have been addressed

Reviewer #3: All comments have been addressed

2. Is the manuscript technically sound, and do the data support the conclusions?

Reviewer #1: Yes

Reviewer #3: Yes

3. Has the statistical analysis been performed appropriately and rigorously? 

Reviewer #1: No

Reviewer #3: Yes

4. Have the authors made all data underlying the findings in their manuscript fully available?

Reviewer #1: Yes

Reviewer #3: Yes

5. Is the manuscript presented in an intelligible fashion and written in standard English?

Reviewer #1: Yes

Reviewer #3: Yes

6. Review Comments to the Author

Reviewer #1: In response to the comment on the inconsistent observation of RAMO cell response to doxorubicin due to Myc down-regulation from literature, the authors provided an argument on different mechanism may be used. Because this is an important issue and it likely will create controversies, the authors are advised to provide an extensive and balanced discussion on this issue, perhaps including discussion not only on Myc but also of eIF3a in drug response.

Reviewer #3: In this revised manuscript, the authors have addressed the main concerns. As a result, the manuscript is suitable for publication.

7. PLOS authors have the option to publish the peer review history of their article (what does this mean?). If published, this will include your full peer review and any attached files.

Reviewer #1: No

Reviewer #3: No

---

## [Author Response · Author response to Decision Letter 1]

8 Sep 2023

Reviewer #1: In response to the comment on the inconsistent observation of RAMO cell response to doxorubicin due to Myc down-regulation from literature, the authors provided an argument on different mechanism may be used. Because this is an important issue and it likely will create controversies, the authors are advised to provide an extensive and balanced discussion on this issue, perhaps including discussion not only on Myc but also of eIF3a in drug response.

As suggested, we have added a short section to the final paragraph of the Discussion noting the conflicting results in the literature (new refs. 46-48). This addition fits well with the following statements that highlight the need for future experiments to better understand EIF3A and MYC roles in different cancer contexts.

Reviewer #3: In this revised manuscript, the authors have addressed the main concerns. As a result, the manuscript is suitable for publication.

We thank the reviewer for the positive feedback.

---

## [Editor Report · Decision Letter 2]

12 Sep 2023

The Helix-Loop-Helix motif of human EIF3A regulates translation of proliferative cellular mRNAs

PONE-D-23-10825R2

Dear Dr. Cate,

We’re pleased to inform you that your manuscript has been judged scientifically suitable for publication and will be formally accepted for publication once it meets all outstanding technical requirements.

Kind regards,

Eric Jan, Ph.D.

Academic Editor

PLOS ONE
---

## [Editor Report · Acceptance letter]

18 Sep 2023

PONE-D-23-10825R2 

The Helix-Loop-Helix motif of human EIF3A regulates translation of proliferative cellular mRNAs 

Dear Dr. Cate:

I'm pleased to inform you that your manuscript has been deemed suitable for publication in PLOS ONE. Congratulations! Your manuscript is now with our production department. 

Kind regards, 

on behalf of

Dr. Eric Jan 

Academic Editor

PLOS ONE